# New Nucleic Base-Tethered Trithiolato-Bridged Dinuclear Ruthenium(II)-Arene Compounds: Synthesis and Antiparasitic Activity

**DOI:** 10.3390/molecules27238173

**Published:** 2022-11-24

**Authors:** Oksana Desiatkina, Martin Mösching, Nicoleta Anghel, Ghalia Boubaker, Yosra Amdouni, Andrew Hemphill, Julien Furrer, Emilia Păunescu

**Affiliations:** 1Department of Chemistry, Biochemistry and Pharmaceutical Sciences, University of Bern, Freiestrasse 3, 3012 Bern, Switzerland; 2Institute of Parasitology, Vetsuisse Faculty, University of Bern, Länggass-Strasse 122, 3012 Bern, Switzerland; 3Laboratoire de Parasitologie, Institution de la Recherche et de l’Enseignement Supérieur Agricoles, Université de la Manouba, École Nationale de Médecine Vétérinaire de Sidi Thabet, Sidi Thabet 2020, Tunisia

**Keywords:** ruthenium(II)-arene complexes, CuAAC reactions, antiparasitic compounds, *Toxoplasma gondii*, human foreskin fibroblasts, toxicity, auxotrophy, nucleic bases

## Abstract

Aiming toward compounds with improved anti-*Toxoplasma* activity by exploiting the parasite auxotrophies, a library of nucleobase-tethered trithiolato-bridged dinuclear ruthenium(II)-arene conjugates was synthesized and evaluated. Structural features such as the type of nucleobase and linking unit were progressively modified. For comparison, diruthenium hybrids with other type of molecules were also synthesized and assessed. A total of 37 compounds (diruthenium conjugates and intermediates) were evaluated in a primary screening for in vitro activity against transgenic *Toxoplasma gondii* tachyzoites constitutively expressing β-galactosidase (*T. gondii* β-gal) at 0.1 and 1 µM. In parallel, the cytotoxicity in non-infected host cells (human foreskin fibroblasts, HFF) was determined by alamarBlue assay. Twenty compounds strongly impairing parasite proliferation with little effect on HFF viability were subjected to *T. gondii* β-gal half maximal inhibitory concentration determination (IC_50_) and their toxicity for HFF was assessed at 2.5 µM. Two promising compounds were identified: **14**, ester conjugate with 9-(2-oxyethyl)adenine, and **36**, a click conjugate bearing a 2-(4-(hydroxymethyl)-1*H*-1,2,3-triazol-1-yl)methyl substituent, with IC_50_ values of 0.059 and 0.111 µM respectively, significantly lower compared to pyrimethamine standard (IC_50_ = 0.326 µM). Both **14** and **36** exhibited low toxicity against HFF when applied at 2.5 µM and are candidates for potential treatment options in a suitable in vivo model.

## 1. Introduction

*Toxoplasma gondii*, the most prevalent infectious protozoan in man, is an opportunistic pathogen with a high zoonotic potential. Approximately 30% of humans worldwide, as well as most warm-blooded animal species, are infected with this parasite [1,2]. Acute toxoplasmosis can be life-threatening in immunocompromised hosts, and upon primary infection during pregnancy, the parasite can cross the placenta and inflict fetal damage and/or abortion [3], rendering *T. gondii* an important public health problem [2,4]. Current management of toxoplasmosis relies on conventional therapy, which has important shortcomings related to reduced tolerance and overall potency, poor efficacy to the latent stage of the parasite, as well as drug resistance [5,6].

As an obligate intracellular pathogen that can invade and replicate in any nucleated mammalian cell, *T. gondii* is in stringent dependence on specific host cell resources [7]. The parasite lacks many genes encoding the entire metabolic pathways, but has evolved efficient strategies to acquire crucial metabolites from the host cells [8]. For survival, *T. gondii* is auxotrophic for numerous nutrients and has acquired salvage mechanisms to import the vital compounds that it cannot synthesize, such as, for example, various metabolites including polyamines, cholesterol, purine derivatives, and essential amino acids [8]. *T. gondii* is a strictly purine (adenine, guanine, xanthine, and hypoxanthine) auxotroph, and is thus incapable of de novo biosynthesis of purine nucleic bases and relies on their scavenging from the host cell to meet its nutritional needs [8,9,10,11,12]. In contrast, *T. gondii* scavenges and synthesizes pyrimidine nucleobases and nucleosides [8,13,14,15,16].

The drugs that are currently used for the treatment of toxoplasmosis include antifolates, such as combined pyrimethamine–sulfadiazine or trimethoprim–sulfamethoxazole. Pyrimethamine can be combined with antibiotics that inhibit protein synthesis such as clindamycin and azithromycin, and atovaquone, a mitochondrial cytochrome *bc1* inhibitor that interferes in oxidative phosphorylation, is also in clinical use. However, these treatments are unspecific, adverse effects have been frequently documented, and clinical failures have been reported [17]. A considerable effort is being made to identify better solutions [17,18,19,20] and new structures are emerging [21,22,23,24,25,26,27]. For example, experimental compounds such as bumped kinase inhibitors, which impair the activity of *T. gondii* calcium dependent protein kinase 1 (essential for host cell invasion and egress), endochin-like quinolones (ELQs) that are specific and potent *T. gondii* cytochrome *bc1* inhibitors, and a leucinostatin-derived antimicrobial peptide that interferes in multipole metabolic pathways, have been demonstrated to exert anti-parasitic effects in vitro and in vivo [28,29,30]. In addition, some combinations of pyrimethamine, clindamycin, guanabenz, and ELQs exhibited synergistic effects and were found to be more potent than monotherapies [31].

Chemotherapeutic strategies addressing the parasitic cell cycle and the specific metabolic defects of *T. gondii* based on its auxotrophic nature were explored earlier [32,33,34]. Targeting the *T. gondii* purine salvage pathways represents a potential pharmacological approach [8,35,36], which was experimentally investigated [37,38,39,40,41,42] but not widely applied due to the apprehension that compounds affecting parasite metabolism can also be toxic to the host [34]. The potential of purine analogues as antiparasitic agents was previously demonstrated [35,43] and several derivatives have shown efficacy against *T. gondii* [44,45].

The research of metal-based compounds for biological applications has received a lot of attention [46,47,48], and if initially most of the bioactive metalorganic compounds were conceived as alternatives to anticancer platinum-based drugs [49,50,51], other pharmacological properties, among which, antiparasitic activity [52,53,54,55,56,57], further encouraged studies in this area. If ruthenium complexes emerged as one of the most promising classes of metal-based anticancer compounds [58,59], they have also shown particularly interesting efficacy against various parasites [60,61,62,63,64,65].

Combining two or more active molecules into a hybrid structure is a popular strategy in the design of new therapeutic agents and prior approaches aiming to improve the targeting and anticancer activity of platinum [66,67] and ruthenium [67] complexes by modifying them with metabolites revealed significant benefits [68]. Conjugating metal complexes to nucleobases, nucleosides, and nucleotides afforded compounds with a wide range of applications, such as, for example, potential anticancer agents [69,70]. An important effort was invested in the development of various metallocene–nucleobase hybrids [70,71,72,73] and research was further extended to other organometallic derivatives [74,75,76,77], some relevant examples being presented in Figure 1.

For example, thymine and adenine ferrocene derivatives **A** and **B** [78,79] and di-ferrocene–uracil compound **C** [80] showed promising anticancer activity and good selectivity. Copper-catalyzed azide-alkyne [3+2] cycloaddition (CuAAC) was used to prepare antitubercular agent **D** [81], anticancer compound **E** [82], and antifungal derivative **F** [72]. CuAAC click reactions also afforded cytostatic bis-ferrocene conjugates **G** bridged by 1,2,3-triazole linkers to 5-substituted uracil [83]. Cymantrene and cyrhetrene uracil and thymine conjugates **H** showed promising activity against Trypanosoma brucei and good selectivity [84], and 5-fluorouracil ruthenium(II)-arene compound I exhibited moderate anticancer activity [85,86]. Thymidine–ruthenium compounds **J**, **K** [87] and **L** [74] exhibited interesting cancer cell toxicity with cellular uptake independent of nucleoside transporters mediation.

Trithiolato-bridged dinuclear ruthenium(II)-arene complexes (e.g., compounds **M**–**O** in Figure 1) are highly active against cancer cells but show limited selectivity [88]. Recent studies revealed that they also exhibit remarkable antiparasitic activity but also interesting selectivity profiles against *Toxoplasma gondii*, *Neospora caninum*, and *Trypanosoma brucei* [89,90,91]. For instance, complex **O** had very low IC_50_ values of 1.2 nM for *T. gondii* and 1 nM for *N. caninum* [89,90].

Trithiolato diruthenium compounds present a structure based on two half-sandwich ruthenium(II)-arene units bridged by three thiols, the Ru_2_S_3_ unit forming a trigonal-bipyramidal framework. The straightforward synthesis and scale-up of mixed trithiolato dinuclear ruthenium complexes like complex **O** (Figure 1) as well as their outstanding chemical inertness make this scaffold a good substrate for functionalization. Diruthenium conjugates with short peptides [92], coumarins [93], BODIPYs [94], and anticancer and antimicrobial drugs [95,96] have been reported and some of them exhibited improved water solubility, as well as enhanced anticancer or antiparasitic activity.

Amide or ester coupling reactions allowed easy modification of mixed trithiolato diruthenium compounds bearing hydroxy, amine, and carboxylic acid groups [93,95,96]. However, as in some cases this type of reaction presented shortcomings [96], other alternatives were envisioned.

The aim of this study was to generate novel hybrid molecules as improved antiparasitic compounds by exploiting the *T. gondii* auxotrophies for nucleic bases. To this end, the synthesis and biological activity assessment of new conjugates nucleobase–trithiolato-bridged dinuclear ruthenium(II)-arene unit were performed. The pool of nucleobases comprised adenine, uracil, cytosine, thymine, and xanthine. Conjugates with other types of small molecules were also synthesized and assessed to ascertain this approach.

Besides the nature of the nucleobase, the influence of the type and length of the linkers between the two moieties were also evaluated. In general, amide or ester coupling reactions allowed easy modification of mixed trithiolato diruthenium compounds bearing hydroxy, amine, and carboxylic acid groups [93,95,96]. However, as in some cases this type of reaction presented shortcomings [96], another objective of this study was to identify and validate new synthetic routes affording hybrids based on the diruthenium unit. The proposed strategy challenged the use of CuAAC reactions [97,98] with the formation of triazole connectors. Azide-alkyne click reactions proved a valuable way for the post-functionalization of metal complexes [81,99,100,101,102,103,104,105,106,107,108,109], whereas the triazole linkers exhibit favorable properties, including a moderate dipole character, hydrogen-bonding capability, rigidity, and stability [110]. Since click reactions require azide and alkyne partners, the use of the trithiolato diruthenium scaffold in either role can be examined in reactions with compounds containing the respective complementary group. A first example of a trithiolato–diruthenium conjugate with metronidazole obtained using a CuAAC reaction was recently reported [96].

Using compound concentrations of 0.1 and 1 µM, the newly obtained hybrids and associated intermediates were submitted to a first in vitro screening, against a transgenic *T. gondii* strain constitutively expressing β-galactosidase (*T. gondii* β-gal) grown in human foreskin fibroblasts (HFF), while their cytotoxicity was evaluated in non-infected HFF by alamarBlue assay. The compounds which at 1 µM exhibited interesting antiparasitic activity (90% tachyzoite proliferation inhibition) and low cytotoxicity (>50% HFF viability), were subjected to a *T. gondii* IC_50_ (half-maximal inhibitory concentration) determination, and HFF cytotoxicity assessment at 2.5 µM.

## 2. Results

### 2.1. Synthesis

#### 2.1.1. Synthesis of the Trithiolato-Bridged Dinuclear Ruthenium(II)-Arene Intermediates **2**–**9**

To access the hybrid molecules, nine diruthenium intermediates **2**–**9** bearing functional groups enabling covalent tethering (i.e., via ester conjugation or click chemistry) of appropriately substituted compounds were first synthesized. Intermediates **2**–**4** functionalized with an amino_,_ carboxy, or hydroxy group, respectively, were obtained following a two-step pathway (Figure 1) using formerly reported procedures [93,111]. Precursor **1**, obtained from the ruthenium dimer ([Ru(*η*^6^-*p*-MeC_6_H_4_Pr*^i^*)Cl_2_]_2_) and (4-(*tert*-butyl)phenyl)methanethiol [93], was reacted with 4-amino-benzenethiol, 2-(4-mercaptophenyl)acetic acid and, respectively, (2-mercaptophenyl)methanol in the appropriate solvent to yield compounds **2**–**4**.

Intermediates **2** and **3** were further reacted with various alkyne derivatives bearing carboxy, hydroxy, and amino groups using ester or amide coupling reactions as presented in Figure 2. Reactions of amino derivative **2** with 5-hexynoic acid and 4-ethynylbenzoic acid (Figure 2, top) in the presence of EDCI (N-(3-dimethylaminopropyl)-N′-ethylcarbodiimide hydrochloride) and HOBt (1-hydroxybenzotriazol) as coupling agents, in basic conditions (DIPEA, N,N-diisopropylethylamine), afforded the ruthenium–alkyne compounds **5** and **6** in 41 and 34% yield, respectively. Alkyne derivatives **7** and **8** were synthetized by reacting carboxy diruthenium compound **3** with propargyl alcohol and propargyl amine (Figure 2, bottom). Ester **7** was obtained in the presence of EDCI as coupling agent and DMAP (4-(dimethylamino)pyridine) as basic catalyst and was isolated in 60% yield, while amide **8** was described previously [96].

The trithiolato diruthenium(II)–arene compounds can also function as an azide partner in click reactions. With this aim, azide-functionalized compound **9** was synthesized following a two-step process (Figure 3 (top)). The hydroxy group of **4** was activated by mesylation with MsCl in basic conditions (TEA, triethylamine), followed by the nucleophilic substitution with azide (NaN_3_); **9** was isolated in 66% yield over two steps.

Diruthenium azide **10** (Figure 3 (bottom)) was obtained by adapting a literature procedure [112], starting from amino derivative **2** using the Sandmeyer reaction. The poor solubility of amine **2** and of the intermediate diazonium salt (not isolated) led to incomplete conversion. Compound **10** was unstable to mild heating and silicagel chromatographic purifications, and only a small quantity of this intermediate was purified for biological evaluation tests. Nevertheless, azide **10** could be used for click reactions even if it contained traces of **2**. Attempts to synthesize **10** using other literature protocols [113,114] were unsuccessful (either the azide was not obtained, or the conversion was lower). Based on their structural features, the various diruthenium conjugates obtained in this study were organized in five families.

#### 2.1.2. Synthesis of Compounds **11**–**13** (Family 1)

In the compounds constituting family 1, the nucleobase moiety was introduced as one of the bridging thiols (Figure 4). Reactions of the dithiolato diruthenium intermediate **1** with 2-thiocytosine, 4-thiouracil, and 2-thioxanthine afforded the mixed trithiolato derivatives **11**, **12**, and **13** in low yields of 13, 23, and 44%, respectively.

The introduction of nucleobase units on the trithiolato diruthenium scaffold using this method presented important limitations. In reactions run with other substrates such as 6-thioguanine, 8-mercaptoadenine, and 2-thiobarbituric acid, the reduced solubility of the thiols in refluxing EtOH led to poor conversions and important difficulties in the recovery of the pure product. Compounds **11** and **12** still contained traces of impurities and their biological activity was not assessed.

#### 2.1.3. Synthesis of Compound **14** (Family 2)

Family 2 comprises ester conjugate **14** obtained by the reaction of carboxy intermediate **3** with 9-(2-hydroxyethyl)adenine **14A** in the presence of EDCI as coupling agent and DMAP as base and was isolated in 43% yield (Figure 5).

#### 2.1.4. Synthesis of Compounds **15**–**26** (Family 3)

The use of the click CuAAC (copper catalyzed azide–alkyne cycloaddition) reactions [101,115] for the obtainment of new conjugates with nucleobases was challenged. A first series of reactions were run using diruthenium azide derivatives **9** and **10** as substrates (Figure 6).

*N*-Propargyl derivatives of uracil (**15**), thymine (**16**), cytosine (**17**), and adenine (**18**) were synthesized by the nucleophilic substitution reactions performed on propargyl bromide with the corresponding nucleobases in basic conditions (K_2_CO_3_), by adapting literature procedures [116,117,118] (Figure 6). Alkyne derivatives **15**–**18** were isolated in low to medium yields (25–48% range). The synthesis of the nucleobase propargyl derivatives **15**–**18** using either directly the nucleobases or appropriately protected intermediates was previously reported [119,120,121]. In this study, the synthesis of derivatives **15**–**18** was not optimized as our purpose was to obtain these intermediates in the quantity and quality allowing the click reactions and the in vitro biological activity screening. The ^1^H and ^13^C NMR data for compounds **15**–**18** (Appendix A) are in good agreement with the published data [122,123,124], taking the different conditions used for the measurements.

The CuAAC reactions were performed by adapting reported protocols [101,115], in the presence of CuSO_4_ as catalyst and sodium ascorbate as reducing agent. Nucleobase propargyl derivatives **15**–**18** were reacted with azides **9** and **10** (Figure 6), affording the click products **19**–**22** and **23**–**26** in low yield (19–48% and 17–30% range, correspondingly). The quantities of isolated products were suitable for a first antiparasitic activity and cytotoxicity screening.

#### 2.1.5. Synthesis of Compounds **27**–**33** (Family 4)

The diruthenium complexes can act as alkyne partner in CuAAC reactions. As azide component, 6-(azidomethyl)uracil **27** was synthesized (isolated 22% yield) by the nucleophilic substitution of the chlorine in 6-(chloromethyl)uracil with NaN_3_ (Figure 7).

The uracil azide **27** was reacted with the alkyne functionalized diruthenium derivatives **5**–**8** (Figure 7) in the presence of CuSO_4_ and sodium ascorbate [101,115], affording the corresponding click products **28**–**31** in low to medium yields (31 to 73% range). These reactions afforded conjugates presenting two types of spacers (aliphatic **28** vs. aromatic **29**) and having ester (**30**) or amide bonds in the linking units (**28**, **29**, and **31**) (Figure 7).

Adenine azide derivative **32** was synthesized in two steps starting from 9-(2-hydroxyethyl)adenine (**14A**) and was reacted with the alkyne functionalized diruthenium compound **8**, to afford conjugate **33** in 13% yield (Figure 8).

#### 2.1.6. Synthesis of Compounds **34**–**39** (Family 5)

To evaluate the impact of the nature of the substituent anchored on the diruthenium scaffold on the biological activity, click conjugates of azides **9** and **10** with various alkyne derivatives such as ethynylbenzene, 4-ethynylbenzyl alcohol, propargyl alcohol, and 2-ethynylpyridine were also synthesized (Figure 9), the reactions being performed in similar conditions [101,115].

If the phenyl derivative **34** was obtained in good yield (90%), the products with polar substituents were isolated only in poor yields (6% for **35** and 19% for **36**). A similar effect was observed in the reactions run with azide **10**, as the phenyl click product **37** was isolated in high yield (79%), while the reactions with more polar ethynyl compounds were less performant affording compounds **38** and **39** only in 26 and, respectively, 9% yield. The alkyne reactivity can increase as the para-substituent becomes more electron-withdrawing, resulting in acidic C_sp_-H bonds [122]. The conversion seems to be influenced also by polar effects not only by substituent electronic properties. This can explain the higher yield obtained with lipophilic phenylacetylene compared to more polar propargyl alcohol and 4-ethynylbenzyl alcohol. Note that the syntheses were not optimized as the aim was to obtain the click compounds satisfying the quantity and quality requirements necessary for a first in vitro bioactivity screening. The reported yields correspond to the isolated pure compounds and do not always reflect the conversion, as the purification of some compounds was more laborious.

All compounds were fully characterized by ^1^H and ^13^C NMR (Nuclear Magnetic Resonance) spectroscopy, ESI-MS (electrospray ionization mass spectrometry) and elemental analysis experiments (the full description is presented in the Supporting information).

#### 2.1.7. Stability of the Compounds

For the biological activity evaluation, 1 mM stock solutions of all compounds were prepared in dimethylsulfoxide (DMSO). 1H-NMR spectra of intermediate 4 and of conjugates **24**, **28**, **33**, and **39** in DMSO-d6, recorded at 25 °C 5 min and 100 days after sample preparation, showed no modifications (see Supporting information), demonstrating very good stability of the diruthenium compounds in this highly complexing solvent.

Furthermore, for evaluating the compounds’ potential nucleobase-pairing via H-bonding interactions, 1H-NMR measurements were performed using DMSO-d6 solutions of uracil, thymine, cytosine, and adenine conjugates **23**, **24**, **25**, and **26** and of the respective complementary nucleic bases (see details in Supporting information). These experiments further evidenced the stability of the diruthenium–nucleobase conjugates in DMSO-d6. The presence of weak H-bonding interactions between the conjugates and the nucleobases was demonstrated by the broadening and the small chemical shift changes of the resonance signals corresponding to the NH groups (Appendix A).

Compounds **7**, **14**, and **30** have an ester linker that can potentially be hydrolyzed in growth media. Comparable conjugates with coumarin and BODIPY fluorescent units linked through ester bonds to the diruthenium unit were recently investigated [93,94]. As a very limited solvolysis of the ester bonds was noticed after 168 h for some compounds, it was concluded that the coumarin and BODIPY diruthenium conjugates exhibit high stability in the conditions used for the biological evaluations [93]. Therefore, it was assumed that compounds **7**, **14**, and **30** are sufficiently stable for the in vitro evaluation.

### 2.2. Assessment of the In Vitro Activity against T. gondii β-gal and Host Cells

#### 2.2.1. Primary Screening

The compounds (conjugates and respective intermediates) were subjected to a sequential biological screening. The antiparasitic activity (proliferation inhibition) was evaluated using *T. gondii* β-gal grown in HFF host cell monolayers and the cytotoxic effects were studied in non-infected HFF monolayers [90]. A first screening (Table 1 and Figure 2) of all compounds against *T. gondii* β-gal tachyzoites and HFF was carried out at concentrations of 0.1 and 1 µM. In a second screening, the selected compounds (which at 1 µM inhibited *T. gondii*-β-gal proliferation by at least 90% and impaired the HFF viability not more than 50%) were submitted to dose-response studies to determine the IC_50_ values, while cytotoxicity in HFF was assessed at 2.5 µM (the results are summarized in Table 2). The same screening strategy was applied in previous studies [111]. Pyrimethamine (IC_50_ = 0.326 µM) was used as reference compound as shown earlier [93].

The antiparasitic activity of compounds **2**–**4** and **8** was previously reported and discussed [93,96,111] and the values are provided in Table 1 and Figure 2 for comparison.

From the diruthenium alkyne intermediates **5**–**8**, only derivative **5** presented reduced toxicity on HFF at 1 µM, while amide **8** affected HFF viability even at 0.1 µM. Compound **5** exhibited an improved antiparasitic efficacy compared to its amino diruthenium precursor **2**. Ester **7** and amide **8** affected more the proliferation of *T. gondii* β-gal than the carboxy precursor **3** but also exerted a stronger effect on the host cells viability at 1 µM. Both diruthenium azides **9** and **10** exhibited high activity against the parasite even at 0.1 µM and were moderately toxic to HFF at 1 µM.

The nucleic base intermediates **14A**, **15**–**18**, **27**, and **32** were non-toxic to the host cells and exhibited no antiparasitic activity against *T. gondii* β-gal at 1 µM.

In family 1, compound **13**, with 2-thioxanthine as one of the thiol bridges, neither affected HFF viability, nor was active against the parasite at both tested concentrations.

The adenine ester conjugate **14** from family 2 exhibited significantly improved antiparasitic activity compared to its precursors **3** and **14A** (9-(2-hydroxyethyl) adenine), almost abolishing parasite proliferation at 1 µM, while exhibiting moderate HFF toxicity at the same concentration.

From the compounds of family 3, the uracil, thymine, cytosine, and adenine click conjugates **19**–**22** were moderately toxic when tested at 1 µM. Nucleobase hybrids **19**–**22** almost abolished parasite proliferation at 1 µM, but only the cytosine conjugate **21** exhibited a strong antiparasitic effect at 0.1 µM.

Compounds **23**–**26** did not affect HFF viability at 0.1 µM, but the uracil and adenine derivatives **23** and **26** presented moderate toxicity to the host cells at 1 µM (HFF viability 76% for both). Among compounds **23**–**26**, the thymine functionalized compound **24** had a stronger antiparasitic effect at 0.1 µM, while cytosine derivative **25** had a low effect on parasite proliferation even at 1 µM. Conjugates **23**, **24**, and **26**, bearing uracil, thymine, and adenine, almost abolished parasite proliferation when applied at 1 µM. Overall, compared to the diruthenium azide precursor **10**, nucleobase click conjugates **23**–**26** applied at 1 µM affected HFF to a lower degree, but exhibited reduced antiparasitic activity at 0.1 µM.

When comparing conjugates presenting the same nucleobase in family 3, cytosine derivatives **21** and **25** exhibited the most striking differences in *T. gondii* β-gal activity. While **21** almost abolished parasite proliferation when applied at 0.1 µM (4%), **25** exerted reduced antiparasitic efficacy even at 1 µM (35%). This suggests that both the linking mode and the type of nucleobase influence the activity of the conjugates.

In family 4, the uracil triazole conjugates **28**–**31** were non-toxic to HFF at 1 µM. However, among their respective diruthenium alkyne intermediates, only **5** did not affect HFF viability at 1 µM, while **6**, **7**, and **8** were toxic to the host cells at the same concentration. Analogues **28**, **29**, and **31**, with amide bonds in their linking unit, presented reduced to no efficacy against *T. gondii* β-gal even at 1 µM. However, compound **30**, with an ester bond in the connecting part, presented a significantly stronger antiparasitic effect compared to amide **31** when applied at 1 µM. Related to uracil derivative **31**, adenine functionalized compound **33** presented a slightly increased activity against the parasite but also higher cytotoxicity to host cells.

In family 5, toxicity differences were observed between click compounds **34**–**36** obtained from the diruthenium azide derivative **9**. Compound **35** showed important HFF toxicity at 1 µM, while derivative **36** was highly active against the parasite at 0.1 µM and was non-toxic to the host cells even at 1 µM. Interestingly, phenyl derivative **37** presented a similar toxicity profile to its analogue **34** with a different mode of linking to the diruthenium moiety. For compounds presenting the methylene–hydroxy group as substituent, **38** exhibited slightly increased HFF toxicity and a reduced antiparasitic efficacy when applied at 0.1 µM compared to **36**. In family 5, compared to precursor diruthenium azides **9** and **10**, only click products **36** and **39** exhibited an improved HFF toxicity/parasite efficacy profile.

Comparing adenine conjugates **14**, **22**, and **26**, the most promising candidate for further studies is the ester analogue **14**. The uracil hybrids **19**, **23**, **28**, **29**, **30**, and **31** exhibited reduced toxicity to HFF at 1 µM but also limited activity against the parasite at 0.1 µM. Compounds **19**, **23**, and **30** presented antiparasitic properties only at 1 µM, while **28**, **29**, and **31** showed low antiparasitic activity even at this concentration.

#### 2.2.2. IC_50_ Values against *T. gondii* β-Gal Tachyzoites and HFF Toxicity at 2.5 µM

In a secondary screening, the dose-response studies (IC_50_ values) against *T. gondii* β-gal tachyzoites and the HFF toxicity at 2.5 µM for 20 selected compounds (which, when applied at 1 µM, inhibited *T. gondii*-β-gal proliferation by at least 90% and did not alter HFF viability by more than 50%) were determined (the results are summarized in Table 2). The same screening strategy was applied in previous studies [111].

Among the diruthenium intermediates, carboxy derivative **3** did not affect HFF viability, amine **2** and propargyl ester **7** exhibited medium toxicity on HFF at 2.5 µM (51% for both), and all the other compounds were highly cytotoxic.

The adenine ester derivative **14** from family 2 was very promising with an IC_50_ value of 0.059 µM (5 times lower than that of the standard drug pyrimethamine, IC_50_ = 0.326 µM), and with moderate effect on HFF cells (76% viability at 2.5 µM, a concentration 40 times higher compared to its IC_50_). Of note, compound **14** exhibited significantly improved antiparasitic activity compared to its diruthenium precursor **3**, with a low increase of host cells cytotoxicity [111].

From family 3, only click conjugates **19**, **22**, **23**, and **24** presented medium or low toxicity on the host cells. Adenine compound **22** exhibited a promising IC_50_ value of 0.108 µM, but also considerably impaired HFF viability (52%). Uracil and thymine click derivatives **23** and **24** had a lower impact on the host cells viability (64 and 85%) but presented IC_50_ values higher or comparable to that of pyrimethamine (0.426 and 0.357 µM). Interestingly, the *ortho* substituted conjugates **19**–**22** are on average slightly more toxic to HFF than the *para* substituted conjugates **23**, **24**, and **26**. Both uracil and thymine click derivatives **23** and **24** are less toxic to HFF than the parent diruthenium azide derivative **10**.

The uracil derivative **30** (click compound presenting an ester bond, family 4) did not affect host cells at 2.5 µM (97%) but exhibited a high IC_50_ value (0.659 µM).

From the click products with other types of substituents (family 5), compounds **34**, **37**, and **39** were highly toxic against HFF at 2.5 µM, and thus, anchoring hydrophobic molecules to the diruthenium scaffold via triazole units appears to be detrimental to the toxicity of HFF. The way in which the substituents are linked to the diruthenium scaffold appears to be important as the triazole–phenyl derivative **34** was significantly more toxic to the host cells compared to its analogue **37** (3 vs. 17%). The triazole–hydroxymethylene compounds **36** and **38** exhibited similar IC_50_ values on *T. gondii* β-gal (0.111 vs. 0.128 µM) for medium toxicity against HFF (77 vs. 59%). Only conjugates **36** and **38** were less toxic to the host cell compared to the parent diruthenium azide derivatives **9** and **10**.

**Table 2 molecules-27-08173-t002:**
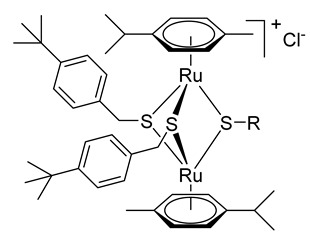
Half maximal inhibitory concentration (IC_50_) values (µM) on *T. gondii* β-gal for the 20 selected compounds and their effect at 2.5 µM on HFF viability.

Compound	R	IC_50_(µM)	[LS; LI] ^c^	SE ^d^	HFF Viability (%) ^e^	SD ^f^
**Diruthenium intermediates**
**4** ^b^	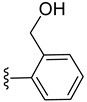	0.038	[0.060; 0.023]	0.110	4	2
**5**	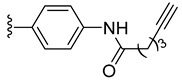	0.038	[0.050; 0.029]	0.063	34	1
**6**	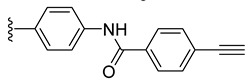	0.288	[0.348; 0.238]	0.188	17	1
**7**	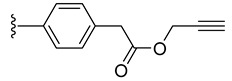	0.289	[0.363; 0.230]	0.229	51	3
**9**	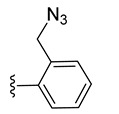	0.048	[0.058; 0.040]	0.139	11	1
**10**	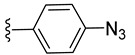	0.064	[0.080; 0.051]	0.050	38	1
**Family 2**
**14**	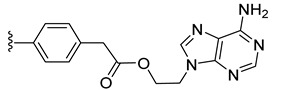	0.059	[0.085; 0.040]	0.037	76	3
**Family 3**
**19**	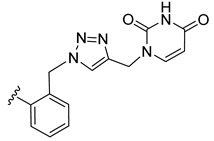	0.460	[0.626; 0.338]	0.307	50	0
**20**	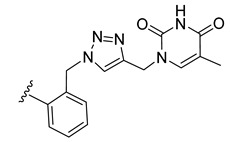	0.363	[0.371; 0.354]	0.023	39	1
**21**	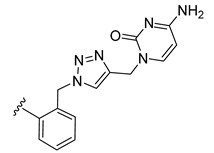	0.046	[0.058; 0.037]	0.048	38	1
**22**	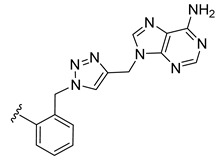	0.108	[0.141; 0.083]	0.066	52	1
**23**	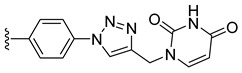	0.426	[0.553; 0.328]	0.260	64	1
**24**	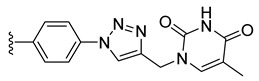	0.357	[0.418; 0.305]	0.156	85	4
**26**	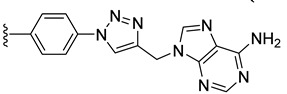	0.178	[0.226; 0.140]	0.061	45	2
**Family 4**
**30**	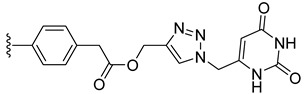	0.659	[0.684; 0.635]	0.037	97	1
**Family 5**
**34**	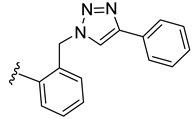	0.092	[0.108; 0.078]	0.038	3	0
**36**	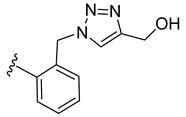	0.111	[0.135; 0.090]	0.039	77	1
**37**	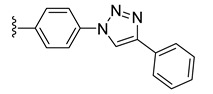	0.096	[0.122; 0.076]	0.057	17	1
**38**	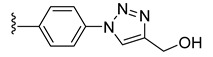	0.128	[0.164; 0.099]	0.058	59	1
**39**	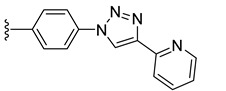	0.075	[0.091; 0.061]	0.043	1	0
**Pyrimethamine** ^a^		0.326	[0.396; 0.288]	0.051	99	6

^a^ Compounds reported in Ref. [93]. ^b^ Compound reported in Ref. [96]. ^c^ Values at 95% confidence interval (CI), LS is the upper limit of CI and LI is the lower limit of CI. ^d^ The standard error of the regression (SE) represents the average distance that the observed values fall from the regression line. ^e^ Control HFF cells treated only with 0.25% DMSO exhibited 100% viability. ^f^ The standard deviation of the mean (six replicate experiments).

No specific SAR (structure–activity relationships) could be identified. Both the attached unit and the connector play an important role in the biological activity. If the interactions of the anchored unit with potential biochemical targets (DNA, proteins) are indeed responsible for the observed biological activity of the conjugates, the structure of some compounds (e.g., those in which the connectors between the two units are longer and more flexible as in conjugates **14**, **28** or **30**, **31** and **33**) could be more favorable compared to that of hybrids in which the access to the pendants is more sterically hindered (e.g., compounds **19**–**22**). The observed biological effects both in terms of host cell toxicity and antiparasitic activity are multifactorial, as along with the attached molecule, the connecting position, the bonds present in the linker and its rigidity, are also very important. Of note, if in the structure of the conjugates the presence of metabolites like nucleobases can favor cell accumulation, the size of the diruthenium unit could constitute a limitative factor in membrane transfer. Assessing the conjugates obtained in this study against other purine auxotrophic parasites, such as *Leishmania donovani* and *Plasmodium falciparum* [123] could further validate the proposed approach.

The most interesting compounds that deserve to be considered for further studies are ester **14** (conjugated to 9-(2-oxyethyl)adenine) and click product **36** (bearing a 2-(4-(hydroxymethyl)-1*H*-1,2,3-triazol-1-yl)methyl substituent). Both **14** and **36** exhibit slightly lower toxicity on HFF compared to the standard drug pyrimethamine but significantly improved antitoxoplasma activity. Adenine ester conjugate **14** exhibits improved antiparasitic properties compared to both its precursors the carboxy diruthenium compound **3** and 9-(2-hydroxyethyl)adenine (**14A**), and this type of connection should be considered for the future development of other nucleobase conjugates.

The mechanisms of action of trithiolato diruthenium compounds have not yet been elucidated. These compounds generally exhibit reduced water solubility [88] and contrary to other Ru(II)-arene complexes presenting labile chlorine or carboxylate ligands, they do not hydrolyze and are stable in the presence of most biomolecules such as amino acids and DNA [88]. Only the oxidation of cysteine (Cys) and glutathione (GSH) to form cystine and GSSG, respectively, was observed in the presence of some compounds, but no correlation between the in vitro cytotoxicity and the catalytic activity on the oxidation reaction of glutathione was observed [124,125]. Recently, inductively coupled plasma mass spectrometry (ICP-MS) experiments proved that complexes **M** and **N** (Figure 1) specifically target the mitochondrion in A2780 ovarian cancer cells [89]. Noteworthy, TEM (transmission electron microscopy) studies of different protozoan parasites (*Toxoplasma gondii*, *Neospora caninum*, *Trypanosoma brucei*) treated with trithiolato dinuclear ruthenium(II)-arene complexes revealed alterations in the mitochondrial ultrastructure pointing out this organelle as a potential target [89]. Interestingly, a similar effect was also observed in the case of trithiolato diruthenium conjugates with coumarin and BODIPY fluorophores [93,94].

The potential cellular and molecular targets of conjugate **14** were recently investigated [126]. Affinity chromatography of parasite extracts on epoxy-sepharose-bound conjugate **14**, which is also highly active against *T. brucei* bloodstream forms [126], showed that this compound interacts with mitochondrial proteins in both parasite species. In *T. gondii*, a major compound **14** binding protein is an homolog to the human mitochondrial import inner membrane translocase subunit Tim1, which is putatively involved in importing proteins from the cytoplasm into the mitochondrion. The mitochondrion as a major target of ruthenium compounds has been identified earlier in both *T. gondii* and *T. brucei,* primarily by TEM [90,91]. However, besides mitochondrial proteins, affinity chromatography has also identified a range of other binding proteins that could be assigned to different cellular pathways, which indicates that this compound could affect several important cellular functions [126]. It is conceivable that, in terms of mode of action, the adenine part of the conjugate **14** could facilitate the uptake of the ruthenium moiety, which would then be processed intracellularly and exert its antiparasitic activity. Further research to clarify the mechanisms that lead to antiparasitic activity is envisaged.

## 3. Materials and Methods

### 3.1. Chemistry

The chemistry experimental part, with full description of procedures and characterization data for all compounds, is presented in the Appendix A.

### 3.2. Biological Evaluation

#### 3.2.1. Cell and Parasite Culture

All tissue culture media were purchased from Gibco-BRL, and biochemical agents were purchased from Sigma-Aldrich. Human foreskin fibroblasts (HFF) were purchased from ATCC, maintained in DMEM (Dulbecco’s Modified Eagle’s Medium) supplemented with 10% fetal calf serum (FCS, Gibco-BRL, Waltham, MA, USA) and antibiotics as previously described [60]. Transgenic *T. gondii* β-gal (expressing the β-galactosidase gene from *Escherichia coli*) were kindly provided by Prof. David Sibley (Washington University, St. Louis, MO, USA) and were maintained, isolated, and prepared for new infections as shown before [60,127].

#### 3.2.2. In Vitro Assessment against *T. gondii* Tachyzoites and Human Foreskin Fibroblasts

The screening cascade for the compounds was described in former studies [111]. All compounds were prepared as 1 mM stock solutions from powder, in dimethyl sulfoxide (DMSO, Sigma, St. Louis, MO, USA). For in vitro activity and cytotoxicity assays, HFF were seeded at 5 × 10^3^/well and allowed to grow to confluence in phenol-red free culture medium at 37 °C and 5% CO_2_. *T. gondii* tachyzoites were released from host cells, and HFF monolayers were infected with freshly isolated parasites (1 × 10^3^/well), and compounds were added concomitantly with infection.

In the primary screening, HFF monolayers infected with *T. gondii* β-gal were treated with 0.1 and 1 µM of each compound, or the corresponding concentration of DMSO (0.01 or 0.1%, respectively) as controls and were incubated for 72 h at 37 °C/5% CO_2_ as previously described [128].

For the dose-response study (IC_50_ values), measurements for *T. gondii* β-gal were performed. The selected compounds were added concomitantly with infection in 8 serial concentrations 0.007, 0.01, 0.03, 0.06, 0.12, 0.25, 0.5, and 1 μM.

To measure the activity of the tested compound on *T. gondii* β-gal parasites, after a period of 72 h of culture at 37 °C/5% CO_2_, the culture medium was aspirated, and cells were permeabilized by adding 90 μL PBS (phosphate buffered saline) with 0.05% Triton X-100. Next, 10 μL of 5 mM chlorophenolred-β-D-galactopyranoside (CPRG; Roche Diagnostics, Rotkreuz, Switzerland, substrate for the β-galactosidase) in PBS were added. Release of chlorophenol red was measured at 570 nm using an EnSpire^®^ multimode plate reader (PerkinElmer, Inc., Waltham, MA, USA). A 100% proliferation of *T. gondii* β-gal was assigned to control parasites treated only with DMSO. For the primary screening at 0.1 and 1 μM, the activity was measured as the release of chlorophenol red over time and was calculated as percentage from the respective DMSO treated control, which represented 100% of *T. gondii* β-gal growth.

For the IC_50_ assays, the activity measured as the release of chlorophenol red over time was proportional to the number of live parasites down to 50 per well as determined in pilot assays. IC_50_ values were calculated after the logit-log-transformation of relative growth and subsequent regression analysis. All calculations were performed using the corresponding software tool contained in the Excel software package (Microsoft, Redmond, WA, USA).

Cytotoxicity assays using uninfected confluent HFF host cells were performed by the alamarBlue assay as previously reported [129]. Confluent HFF monolayers in 96 well-plates were exposed to 0.1, 1, and 2.5 μM of each compound. Non-treated HFF as well as DMSO controls (0.01%, 0.1%, and 0.25%) were included. After 72 h of incubation at 37 °C/5% CO_2_, the medium was removed, and the plates were washed once with PBS. Resazurin suspended in PBS to a final concentration of 0.01 g/L was added to each well (200 μL/well). Fluorescence was measured at excitation wavelength 530 nm and emission wavelength 590 nM using an EnSpire^®^ multimode plate reader (PerkinElmer, Inc, Waltham, MA, USA). The cytotoxicity values were calculated as percentage of the respective DMSO control, which represented 100% of HFF viability.

## 4. Conclusions

This study was focused on the synthesis and antiparasitic activity assessment of nucleobase-tethered trithiolato-bridged dinuclear ruthenium(II)-arene compounds, aiming at exploiting the parasite auxotrophies and metabolic peculiarities for nucleic bases. Various synthetic strategies and structural modifications were considered, affording 23 new diruthenium conjugates. The CuAAC reactions proved to be a convenient method for the functionalization of trithiolato diruthenium compounds with various substrates including nucleobase derivatives. The organometallic fragment was used both as alkyne and azide group bearing partner and allowed the isolation of 19 new click conjugates. The conjugate synthesis based on CuAAC reactions can be used to develop other thiolato-bridged ruthenium(II)-arene hybrids in a convenient manner and to construct larger libraries of compounds.

The antiparasitic activity and the toxicity on the host cells were influenced not only by the nature of the molecule appended to the diruthenium scaffold but also by the linker between the two moieties. Even though some of the compounds presented in this study exhibit good antitoxoplasma activity and low HFF toxicity, when considering the overall results, targeting *T. gondii* metabolic pathways using nucleobase conjugates of trithiolato diruthenium derivatives does not appear to be a promising approach as the hybrid molecules presenting nucleic base units did not necessarily performed better compared to conjugates presenting other types of substituents. However, further evaluations of the conjugates against other nucleobase auxotrophic parasites, such as *Leishmania donovani* and *Plasmodium* should be performed to conclusively appraise the proposed strategy.

Ester **14**, conjugated to 9-(2-oxyethyl)adenine, and click product **36**, bearing a 2-(4-(hydroxymethyl)-1*H*-1,2,3-triazol-1-yl)methyl substituent, exhibited interesting IC_50_ values on *T. gondii* β-gal of 0.059 and 0.111 µM, respectively, with only medium toxicity to HFF at 2.5 µM. These two hybrid molecules deserve further investigations.

## Data Availability

The data are included in the article and Appendix A.

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
