# Peer review of "New Nucleic Base-Tethered Trithiolato-Bridged Dinuclear Ruthenium(II)-Arene Compounds: Synthesis and Antiparasitic Activity"

_molecules, 2022, doi:10.3390/molecules27238173_

Round 1

Reviewer 1 Report

The Manuscript by O. Desyatkina, M. Mösching, N. Anghel, G. Boubaker, Y. Amdouni, A. Hemphill, J. Furrer, E. Păunescu “New Nucleic Base-Tethered Trithiolato-bridged Dinuclear Ruthenium(II)-Arene Compounds: Synthesis and Antiparasitic Activity” describes the synthesis of a wide range of trithiolato-bridged ruthenium(II)-arene derivatives with potential antiparasitic activity and represents a continuation of scientific group’s research aiming at the development of novel bioactive organometallic compounds. High stability, high antitoxoplasma activity and moderate toxicity of thiolato-bridged ruthenium complexes allows considering them as promising agents for the treatment of toxoplasmosis and other infectious diseases. In the manuscript, methods for the synthesis of novel derivatives of ruthenium complexes, including natural nucleobases derivatives, are proposed. Despite the complications associated with limited reactivity and solubility of natural purine and pyrimidine derivatives, authors succeeded in the synthesis of a broad number of hybrid molecules and assessed their cytotoxicity and activity against Toxoplasma gondii using β-galactosidase-producing T. gondii strain. The synthesis of new compounds is described well, the compounds’ structure and results of biological testing are beyond any doubts. However, there are still some questions and moments to be clarified.

1.    In Introduction, the articles published not later than in 2019 are cited. The actuality of the development of novel antitoxoplasma agents should be emphasized by citing more recent relevant publications.

2.    In the part of synthesis of compounds of Family 3, the preparation of starting propargyl derivatives 15-18 has to be discussed in more detail. If alkylation of pyrimidine nucleobases leads exclusively to 1-substituted compounds 15-17, it should be mentioned in the text of paper or at least in Supporting information. Thus, the articles [102-104] cited in the manuscript describe that amino group in position 4 of cytosine has to be protected prior to alkylation step. It is necessary to cite the articles decribing alkylation of unsubstituted cytosine with propargyl bromide (e.g. [New J. Chem. 2010, 34, 2634-2644; Chin. Chem. Lett. 2021, 32, 2479-2483]) and to compare physicochemical properties of compounds 15-18 (melting point, NMR spectra) with published data ([Tetrahedron 2000, 56, 1233-1245; Arkivoc 2009, (xiii), 142-152 etc.].

3.    The yields of Cu-catalyzed cycloaddition reactions involving alkynyl derivatives 5 and 6 containing aliphatic and aromatic spacers respectively (Scheme 7) differ significantly. The yields of phenylacetylene derivatives 34 and 37 are also the highest ones (Scheme 9). If CuAACs commonly proceed in higher yield in the case of arylacetylenes and in lower yield in the case of linear terminal alkynes, some comments in the text of paper are desirable.

4.    In Table 1, the meaning of columns “T. gondii β-gal growth (%)” have to be formulated more clearly. I propose to clarify this index either in the head or in the footnote of Table 1. Moreover, it’s desirable to add some reference compound in Table 1 and make appropriate corrections in the text.

5.    Discussion of the possible mechanism of action of compounds on T. gondii repeats the abstract of authors’ publication in Int. J. Mol. Sci. I think, it would be better to discuss of the mechanism of action of leading compound in more detail taking in account the targets and mechanism of action of known ruthenium-organic compounds.

I believe that the Paper by by O. Desyatkina, M. Mösching, N. Anghel and co-authors is interesting for the broad auditory of the Molecules journal and can be published after corrections mentioned above.

Author Response

The Manuscript by O. Desyatkina, M. Mösching, N. Anghel, G. Boubaker, Y. Amdouni, A. Hemphill, J. Furrer, E. Păunescu “New Nucleic Base-Tethered Trithiolato-bridged Dinuclear Ruthenium(II)-Arene Compounds: Synthesis and Antiparasitic Activity” describes the synthesis of a wide range of trithiolato-bridged ruthenium(II)-arene derivatives with potential antiparasitic activity and represents a continuation of scientific group’s research aiming at the development of novel bioactive organometallic compounds. High stability, high antitoxoplasma activity and moderate toxicity of thiolato-bridged ruthenium complexes allows considering them as promising agents for the treatment of toxoplasmosis and other infectious diseases. In the manuscript, methods for the synthesis of novel derivatives of ruthenium complexes, including natural nucleobases derivatives, are proposed. Despite the complications associated with limited reactivity and solubility of natural purine and pyrimidine derivatives, authors succeeded in the synthesis of a broad number of hybrid molecules and assessed their cytotoxicity and activity against Toxoplasma gondii using β-galactosidase-producing T. gondii strain. The synthesis of new compounds is described well, the compounds’ structure and results of biological testing are beyond any doubts. However, there are still some questions and moments to be clarified.

  1. In Introduction, the articles published not later than in 2019 are cited. The actuality of the development of novel antitoxoplasma agents should be emphasized by citing more recent relevant publications.

  • This is absolutely correct, and we thank this referee for pointing out this lack. We have therefore added several recent references. A small paragraph was introduced in the text of the manuscript – lines 61-70, page 2.

  1. In the part of synthesis of compounds of Family 3, the preparation of starting propargyl derivatives 15-18 has to be discussed in more detail. If alkylation of pyrimidine nucleobases leads exclusively to 1-substituted compounds 15-17, it should be mentioned in the text of paper or at least in Supporting information. Thus, the articles [102-104] cited in the manuscript describe that amino group in position 4 of cytosine has to be protected prior to alkylation step. It is necessary to cite the articles decribing alkylation of unsubstituted cytosine with propargyl bromide (e.g. [New J. Chem. 2010, 34, 2634-2644; Chin. Chem. Lett. 2021, 32, 2479-2483]) and to compare physicochemical properties of compounds 15-18 (melting point, NMR spectra) with published data ([Tetrahedron 2000, 56, 1233-1245; Arkivoc 2009, (xiii), 142-152 etc.].

  • Additional explanations accompanied by relevant references were introduced in the text of the manuscript – lines 249-257, page 8. Note that in reference 103, the preparation of monopropargylated nucleobases is described as follows: “uracil, thymine, 6-azauracil and adenine were used as starting materials that were treated with propargylbromide in the presence of K2CO3. All reactions were carried out in DMF, as it is an excellent solvent for dissolving nucleobases [1]. The pyrimidine and as-triazine derivatives were exclusively alkylated at the N-1 position, and the purine in N-9 position, as confirmed by 1H-NMR and 13C-NMR spectra. [1]: 2009;xiii:142–152. Suggested by the referee.

  1. The yields of Cu-catalyzed cycloaddition reactions involving alkynyl derivatives 5 and 6 containing aliphatic and aromatic spacers respectively (Scheme 7) differ significantly. The yields of phenylacetylene derivatives 34 and 37 are also the highest ones (Scheme 9). If CuAACs commonly proceed in higher yield in the case of arylacetylenes and in lower yield in the case of linear terminal alkynes, some comments in the text of paper are desirable.

  • Additional comments were introduced in the text of the manuscript – lines 300-308, page 10.

  1. In Table 1, the meaning of columns “T. gondii β-gal growth (%)” have to be formulated more clearly. I propose to clarify this index either in the head or in the footnote of Table 1. Moreover, it’s desirable to add some reference compound in Table 1 and make appropriate corrections in the text.

  • We agree that for non-specialists, this description may be unclear. Comments were introduced in the footnote of Table 1 – lines 357-358, page 14, and lines 347-348 page 11. Table 1 shows the results of the initial screening, where compounds are tested at 0.1 and 1 mM concentrations. The results are used to select compounds that meet our 2 selection criteria, as mentioned in the text (at 1 µM, inhibition of gondii-β-gal proliferation by at least 90% and impairment of the HFF viability by not more than 50%), and for which the IC50 and toxicity to HFF are determined (Table 2). For this second evaluation, a reference drug, pyrimethamine, is included in the results. We therefore believe that it is not necessary to include this reference compound for the first evaluation, nor have we ever done so in our previous studies.
  1. Discussion of the possible mechanism of action of compounds on T. gondii repeats the abstract of authors’ publication in Int. J. Mol. Sci. I think, it would be better to discuss of the mechanism of action of leading compound in more detail taking in account the targets and mechanism of action of known ruthenium-organic compounds.

  • We thank this referee for his comment and suggestion. The main problem with our compounds is that precisely the target(s) and mechanism of action(s) remain unknown to a significant extent. A small paragraph with comments and appropriate references was introduced in the text of the manuscript – lines 502-532, pages 18-19.

Reviewer 2 Report

Desiatkina et al. submitted the manuscript "New Nucleic Base-Tethered Trithiolato-bridged Dinuclear Ruthenium(II)-Arene Compounds: Synthesis and Antiparasitic Activity" under Section "Inorganic Chemistry" and special issue "Metal Complexes as Potential Antimicrobial and Antiproliferative Agents" is about the development of antiprotozoal agents (Toxoplasma Gondii) based on metal-organic compounds/frames.

The strength of the paper is the synthesis and comprehensive chemical characterization of compounds.  

Minor comments

1.                        Please improve the rationality of Ru-based compounds as antiprotozoals-

(a)                     Protozoa are distinctive when compared to each other in terms of intracellular and extracellular components; what are the targets/target of these synthetics in Toxoplasma Gondii, are not adequately explained.

(b)                     Although authors show potent-to-moderate antitaxoplasma activity in their study for these synthetics, I believe there is also a role in the extent of compound solubility. For example, organic-metallic compounds have inherited solubility problems; therefore, an aberrant activity can occasionally be observed. If authors observed such solubility issues with any of these compounds, then that must be included in the paper.

(c)                      The arrangement of organic group ligands and conformation of these ligands in the metallic organic frame plays a crucial role in indicating a potent activity for metal-based organic compounds. Please add a small paragraph indicating the importance of such points, which will enhance the paper's readability.

2.                        There is no clear indication of control used by the author in their study; please state clearly if used.

3.                        To be a successful medicinal chemistry paper, this manuscript needs to elaborate on the structure-activity relationship (SAR) of these synthetics. However, the authors discussed the SAR of some of the compounds.

The manuscript is well-written and systematically organized and can be considered for the current journal if the authors revise it based on the abovementioned points.

Author Response

Desiatkina et al. submitted the manuscript "New Nucleic Base-Tethered Trithiolato-bridged Dinuclear Ruthenium(II)-Arene Compounds: Synthesis and Antiparasitic Activity" under Section "Inorganic Chemistry" and special issue "Metal Complexes as Potential Antimicrobial and Antiproliferative Agents" is about the development of antiprotozoal agents (Toxoplasma Gondii) based on metal-organic compounds/frames.

The strength of the paper is the synthesis and comprehensive chemical characterization of compounds.  

Minor comments

  1. Please improve the rationality of Ru-based compounds as antiprotozoals-

  • Additional appropriate references sustaining the interest of ruthetnium-based compounds as potential antiparasitic active compounds were introduced in the text of the manuscript – line 84, page 2.

  • Protozoa are distinctive when compared to each other in terms of intracellular and extracellular components; what are the targets/target of these synthetics in Toxoplasma Gondii, are not adequately explained.

  • The rationale of this study is explained in the paragraph ‘Introduction’ of the manuscript - lines 44-54, page 2, as well as lines 126-131, page 4.

(b)                     Although authors show potent-to-moderate antitaxoplasma activity in their study for these synthetics, I believe there is also a role in the extent of compound solubility. For example, organic-metallic compounds have inherited solubility problems; therefore, an aberrant activity can occasionally be observed. If authors observed such solubility issues with any of these compounds, then that must be included in the paper.

  • No solubility issues were observed in the conditions used for the performed in vitro tests. As mentioned in the Material and Methods section (p19, lines 548-551), All compounds were prepared as 1 mM stock solutions from powder, in dimethyl sulfoxide, in which the solubility is good.

(c)                      The arrangement of organic group ligands and conformation of these ligands in the metallic organic frame plays a crucial role in indicating a potent activity for metal-based organic compounds. Please add a small paragraph indicating the importance of such points, which will enhance the paper's readability.

  • The metal complexes in this study present two identical bridging thiols and thus an increased degree of symmetry (no conformational isomers). In the conjugates none of the units appended on one the bridging thiols contains chiral centers.

Additional comments were introduced in lines 480-490, page 18.

  1. There is no clear indication of control used by the author in their study; please state clearly if used.

  • Pyrimethamine has been used as control & standard drug. This is mentioned in the text, table 2 & abstract (lines 357-358, page 14, and lines 347-348 page 11). In footnote of Table 2 and in the Materials and methods section (point 3.2.2.), it is mentioned that Control HFF cells treated only with 0.25% DMSO exhibited 100% viability”.

  1. To be a successful medicinal chemistry paper, this manuscript needs to elaborate on the structure-activity relationship (SAR) of these synthetics. However, the authors discussed the SAR of some of the compounds.

  • This is a matter of fact that we could not find any evidence of clear SAR, as mentioned in the text lines 479-480, page 18. ‘No specific SAR (structure-activity relationships) could be identified. Both the attached unit and the connector play an important role in the biological activity.’ As there are many parameters that can influence the biological activity, a complete SAR study requires the synthesis and biological activity screening of a much larger compound library, with step-by-step structural modifications. Our aim with this study was not to make a SAR study but to validate the concept that exploiting the gondii auxotrophies for nucleic bases could be a strategy for the obtainment of anti-Toxoplasma compounds with improved antiparasitic activity and selectivity, meaning that at least some of the new conjugates nucleobase-diruthenium complex exhibit a better antiparasitic efficacy/cytotoxicity balance compared to the parent diruthenium compound which was proved in the case of adenine conjugate 14. Additional comments were introduced in lines 480-490, page 18.